# Berberine Photo-Activation Potentiates Cytotoxicity in Human Astrocytoma Cells through Apoptosis Induction

**DOI:** 10.3390/jpm11100942

**Published:** 2021-09-22

**Authors:** Francesca Carriero, Carolina Martinelli, Fabio Gabriele, Giulia Barbieri, Lisa Zanoletti, Gloria Milanesi, Claudio Casali, Alberto Azzalin, Federico Manai, Mayra Paolillo, Sergio Comincini

**Affiliations:** 1Department of Biology and Biotechnology, University of Pavia, 27100 Pavia, Italy; francesca.carriero01@universitadipavia.it (F.C.); carolina.martinelli01@universitadipavia.it (C.M.); fgabriele.90@gmail.com (F.G.); giulia.barbieri@unipv.it (G.B.); lisa.zanoletti01@universitadipavia.it (L.Z.); gloria.milanesi@unipv.it (G.M.); claudio.casali01@universitadipavia.it (C.C.); alberto.azzalin@unipv.it (A.A.); federico.manai01@universitadipavia.it (F.M.); 2SKYTEC Srl, 20147 Milan, Italy; 3Department of Drug Science, University of Pavia, 27100 Pavia, Italy; mayra.paolillo@unipv.it

**Keywords:** oxidative stress, phyto-therapy, astrocytoma, glial tumor

## Abstract

Photodynamic therapy (PDT) has recently attracted interest as an innovative and adjuvant treatment for different cancers including malignant gliomas. Among these, Glioblastoma (GBM) is the most prevalent neoplasm in the central nervous system. Despite conventional therapeutic approaches that include surgical removal, radiation, and chemotherapy, GBM is characterized by an extremely poor prognosis and a high rate of recurrence. PDT is a physical process that induces tumor cell death through the genesis and accumulation of reactive oxygen species (ROS) produced by light energy interaction with a photosensitizing agent. In this contribution, we explored the potentiality of the plant alkaloid berberine (BBR) as a photosensitizing and cytotoxic agent coupled with a PDT scheme using a blue light source in human established astrocytoma cell lines. Our data mainly indicated for the combined BBR-PDT scheme a potent activation of the apoptosis pathway, through a massive ROS production, a great extent of mitochondria depolarization, and the sub-sequent activation of caspases. Altogether, these results demonstrated that BBR is an efficient photosensitizer agent and that its association with PDT may be a potential anticancer strategy for high malignant gliomas.

## 1. Introduction

Glioblastoma (GBM, WHO grade IV) is the most common and aggressive type of malignant primary brain tumor in humans [1]. When adopting the current treatment options, i.e., surgery tumor ablation, chemo- and/or radio-therapy protocols, relatively unchanged in the last decade [2], the disease inevitably progresses, however, and relapses within few months post diagnosis [3]. Despite a multitude of research efforts, ongoing therapeutic strategies, including for example first-line chemotherapeutic compounds as temozolomide or carmustine, have not determined a significantly higher efficacy in clinics mostly due to limited pharmacokinetics and to GBM intrinsic and acquired resistance to the treatments [4]. In this regard, the large extent of genetic and phenotypic heterogeneity among GBMs [5] has constituted a biological hurdle for the efficacy of the therapeutic agents toward tumor cells [6]. Therefore, exploiting new strategies that might overcome the cell resistance and heterogeneity features will be crucial to improve the quality of life and survival time of GBM patients [7]. Recently, clinical applications of photodynamic therapy (PDT) in oncologic contexts including GBM have gained increased attention [8,9,10]. PDT generates reactive oxygen species (ROS) in the target tissue by a combination of light, photosensitizers (PS), and oxygen, causing a massive oxidative stress within the tumor cells. In turn, ROS rapidly accumulate and interact with macromolecules as proteins, unsaturated fatty, acids and cholesterols, irreversibly damaging the integrity and functionality of the membranes of intracellular organelles, particularly mitochondria, lysosomes, and the endoplasmic reticulum, ultimately inducing irreversible cell death fates [11]. The efficacy of ROS-induced photo-toxic effects depends on the uptake of PS within cancer cells and on their specific intracellular localization. To date, different clinical trials based on a growing list of PS as hematoporphyrin derivative (commercially known as Photofrin), talaporfin sodium (Laserphyrin), 5-ALA (Gliolan), and meta-tetra(hydroxyphenyl) chlorin (Foscan), gave significant improvement in the performance of fluorescence-guided surgery, with however relatively modest effectiveness for the prognosis of GBM patients [12,13].

Recent progresses in the functional screening of putative anticancer agents relate to “natural bio-active(s)” molecules, mostly isolated from plants, namely quercetin, curcumin, and berberine (BBR). These compounds are generally characterized by intrinsic fluorescence spectra and accompanied by a reduced toxicity toward normal cells [14,15]. In particular, BBR isolated from *Rhizoma coptidis* showed a broad spectrum of antitumor activity even in GBM, through the activation of the apoptosis and autophagy processes, thus producing significant in vitro and in vivo reduction of proliferation and migration indexes [15]. However, the efficacy of BBR is relatively limited due to a reduced oral bioavailability (approximately less than 5%) and to efflux effects produced by P-glycoproteins activity at the cancer cell membranes [16]. Importantly, it was reported that BBR is an effective photosensitizing agent in PDT schemes, through its ability to induce massive ROS intracellular production in response to light energy stimulations [17]. Inbaraj et al. [18] reported in vitro cytotoxicity of BBR upon UVA irradiation in HaCaT immortalized human keratinocytes, while, in a different study, the blue light irradiation of BBR proved to be effective in reducing the viability of U87-MG GBM cell line [19]. Notably, BBR can selectively and preferentially accumulate into the mitochondria of cancer cells, due to its hydrophobicity and cationic surface charges [20]. In general, the specific sub-cellular localization of a PS is crucial to address in a defined programmed type of cell death; indeed, apoptosis is triggered when PS is stored preferentially within the mitochondria organelles. In this regard, in different tumors, BBR has been reported to induce an intrinsic apoptosis pathway, characterized by alterations in the mitochondrial membrane potential, accumulation of ROS, and finally to overall changes in permeability of mitochondrial membranes [21].

Although to date, there are different contributions documenting the effectiveness of the combined in vitro BBR-photo-activation schemes, only a few are directly focused on the molecular mechanisms evoked in GBM cells. Therefore, this contribution focused on the development of an in vitro PDT protocol based on the effect of BBR in GBM cells, highlighting the molecular pathways implicated in cell death induction.

## 2. Materials and Methods

### 2.1. Cell Culture and Chemicals

High grade astrocytoma established cell lines (i.e., T98G, U87-MG, U373-MG, and U138-MG) were obtained from the American Type Culture Collection (Manassas, VA, USA); Res186 low grade pediatric astrocytoma and rat normal astrocytes were respectively provided by Dr. M. Bobola (University of Washington, Seattle, WA, USA) and Prof. S. Schinelli (University of Pavia, Italy) and respectively described in [22,23]. Cells were routinely grown as monolayers at 37 °C in Dulbecco’s modified Eagle’s medium (DMEM) supplemented with 10% fetal bovine serum (FBS) and 100 U/mL of penicillin and 100 μg/mL of penicillin-streptomycin (all reagents from Euroclone, Milan, Italy), under atmosphere controlled at 5% CO_2_.

Berberine (BBR, 99% purity, Sigma-Aldrich, St. Louis, MO, USA) was resuspended in dimethyl sulfoxide (DMSO) and administered at different concentrations (i.e., 20, 50, 100, 200 μg/mL) for 4 h. Next, cells were washed with PBS and incubated for 24 h post treatment (p.t.) with complete media without BBR before trypsinization. Control-mock administration experiments (using the same DMSO concentration adopted for BBR solutions) were performed to exclude DMSO-induced cytotoxicity (data not shown). To stain mitochondria in living cells, MitoTracker Deep Red FM (Thermofisher, Waltham, MA, USA), a far red-fluorescent dye (abs/em 644/665 nm) was employed as follows: cells were incubated with 10 nM dye at 37 °C for 45 min and then visualized by inverted fluorescent and confocal microscopes or trypsinized for cytofluorimetric analysis. The pan-caspase inhibitor Z-VAD(OMe)-FMK, purchased from Cell Signaling (Danvers, MA, USA), was used at 100 μM for 4 h. After this time interval, media were removed following PBS washing and fresh media without BBR/Z-VAD(OMe)-FMK was provided. Finally, cells were evaluated by cytofluorimetric analysis after 24 h p.t. To evaluate differences in the autophagy flux, the expression of LC3B-II isoform was assayed following bafilomycin A1 (Sigma-Aldrich) administration (10 nM at 4 h p.t.) and finally protein expression visualized by immunoblotting.

### 2.2. Cell Viability Assays

Cells were seeded at a density of 8 × 10^3^ cells/well in 96-well plates in a volume of 200 μL for 24 h and then treated with several concentrations of BBR (20, 50, 100, 200 μg/mL for 4 h). After 24 h p.t., 20 μL of Cell Titer One Aqueous Solution (Promega, Madison, WI, USA) was added in each well and incubated for 2 h at 37 °C. Then, absorbance was measured using a microplate reader (Sunrise, Tecan, Männedorf, Switzerland) at a wavelength of 492 nm. All experiments were performed in triplicate with independent assays.

### 2.3. BBR Photo-Stimulation

Cells were seeded at a density of 10^5^ cells in 30 mm plates for 24 h. Cells (with an approximate 80% confluence) were then incubated for 4 h with BBR (200 μg/mL). Four treatment groups were considered: Group 1 (control, untreated), Group 2 (LED-irradiated control), Group 3 (BBR-only incubation), Group 4 (BBR + LED, incubated for 4 h with BBR and irradiated). After BBR incubation, cells were washed with PBS and 1 mL of fresh culture medium was then added to each plate. A blue LED source (Safe Imager 2.0 Blue-Light Transilluminator, Invitrogen, Carlsbad, CA, USA), operating at 447 nm and 1.2 mW/cm^2^ of intensity, set for 4 min of application, was used for irradiation. All experiments were performed in triplicate independent assays.

### 2.4. Muse Cytofluorimetric Assays

A Muse cytofluorimeter and the corresponding assays (Luminex, Austin, TX, USA) were adopted for analysis of Annexin V (Annexin V and Cell Dead kit), caspases activation (Caspase 3/7, and MultiCaspase kits), evaluation of Reactive Oxygen Species (ROS) (Oxydative Stress kit), mitochondria depolarization (MitoPotential kit), autophagy activation (Autophagy LC3-Antibody Based kit) in T98G cells as untreated (NT), exposed to led-only stimulation (LED), BBR (200 μg/mL for 4 h) administration and with the combined LED+BBR scheme, all evaluated after 24 h p.t. For cytofluorimetric evaluations, trypsinized cells, derived from 10^5^ cells in 30 mm plates grown for 24 h with an approximate 80% confluence, were analyzed according to the manufacturer’s specifications of each assay. Experiments were performed in duplicates or triplicates, evaluating 2000 cells for each cytofluorimetric assay.

### 2.5. Imagestream Flow Cytometry Analysis

An Amnis ImageStream MkII instrument (Luminex) equipped with 3 lasers (100 mW 488 nm, 150 mW 642 nm, 70 mW 785 nm (SSC) was used to assay BBR cellular internalization, BBR and mitochondria co-localization, nuclear fragmentation, and Cytocrome c expression. For the analysis of BBR internalization, cells were incubated in the presence of 200 μg/mL of compound for 4 h. Growth medium was then replaced with BBR-free, fresh growth medium and, after 24 h p.t., cells were trypsinized and analyzed by flow cytometry at 60× magnification (NA = 0.9; DOF = 2.5 μm, core size = 7 μm), excited at 488 nm (Ch02, 480–560 nm, Ch width, 528/65 bandpass, at 10 mW). For nuclear staining, before trypsinization, cells were incubated at room temperature for 15 min with DRAQ5 nuclear dye (Invitrogen, 1 μL of a 1:10 dilution/30 mm plate) and analyzed by flow cytometry at 60× magnification, excited at 642 nm (Ch05, 642–745 nm, Ch width, 762/35 bandpass). Mito Tracker (Invitrogen) marker was used to highlight mitochondria, exciting with 642 nm laser. Cytochrome c antibody (#D18C7, Cell Signaling) was conjugated with DyLight 650 labelling kit (Biorad) and cytofluorimetric assayed. Channel 6 (745–800 nm filter) was used for scatterplot (SSC) detection and standard sheath fluid (D-PBS, Themofisher) was adopted in all measurements. Data acquisition was performed by INSPIRE software v0.3, while data analysis using the following IDEAS v6.2 (Luminex) wizard tools: Internalization, Nuclear localization, Spot counting according to the gating specifications.

### 2.6. Microscopy Analysis

T98G cells were seeded into 35 mm glass-bottom petri dishes at a density of 10^4^/mL, and allowed to recover for 24 h prior to analysis.

Phase contrast and fluorescence images were obtained using an inverted microscope (Nikon Eclipse TS100) using a 40× and a 100× plan fluor oil immersion objective, respectively as described [24]. For BBR fluorescence detection, the UV2A (Nikon, Tokyo, Japan) block filter (355/50 nm; dichroic mirror = 400, barrier filter = 410) was employed. BBR and mitochondria were also visualized using a Nanolive 3D CELL EXPLORER-fluo microscope (MEDIA System Lab, Macherio, Italy) using fluorescence channels FITC (excitation peak at 491 nm and an emission peak at 516 nm) and Cy5 (excitation peak at 646 nm and an emission peak at 664 nm), respectively for BBR and MitoTracker.

For confocal microscopy, BBR and mitochondria fluorescence were imaged using a TCS SP8 X system (Leica, Wetzlar, Germany) with excitation peak at 488 nm and emission at 500–600 nm for BBR and excitation peak at 641 nm and emission at 650–750 nm for MitoTracker, with a 63X oil objective (NA = 1,4, zoom 2).

For transmission electron microscopy (TEM) analysis, T98G cells (10^6^) were grown in DMEM medium in 90 mm plates. Following LED and/or BBR (200 μg/mL for 4 h) stimulation for 4 min, after additional 24 h p.t., cells were harvested by centrifugation at 800 rpm for 5 min and fixed with 2.5% glutaraldehyde in PBS, for 2 h at room temperature as described [25]. Cells were then rinsed in PBS (pH 7.2) overnight and post-fixed in 1% aqueous OsO_4_ for 1 h at room temperature. Cells were pre-embedded in 2% agarose in water, dehydrated in acetone, and finally embedded in epoxy resin (Electron Microscopy Sciences, EM-bed812). Ultrathin sections (60–80 nm) were collected on nickel grids and stained with uranyl acetate and lead citrate. The specimens were observed with a JEM 1200 EX II (JEOL, Peabody, MA, USA) electron microscope, equipped with a MegaView G2 CCD camera (Olympus OSIS, Tokyo, Japan) and operating at 120 kV. The morphology of mitochondria (at least 20 for each sample) was then analyzed by two independent evaluators.

### 2.7. DNA Extraction and Electrophoretic Analysis

Genomic DNA was isolated from 10^6^ T98G cells (i.e., untreated; LED-only; BBR 200 μg/mL for 4 h; BBR combined with led stimulation) as follows: cell pellets were gently resuspended into 50 μL of TES lysis buffer (Tris-OH pH 7.5 10 mM; EDTA pH 8.0, 10 mM; SDS 0.5% (*v*/*v)*, all from Sigma-Aldrich), supplemented with 10 μL of RNAse Cocktail (Thermofisher), mixed well by flipping the tip of the tubes without vortexing and incubating for 90 min at 37 °C. Then, 10 μL of proteinase K (Sigma-Aldrich) was added following an incubation of 50 °C for 90 min. DNA was then quantified by a Qubit fluorimeter using DNA HS Assay kit (Thermofisher) and finally loaded into 2% (*w*/*v*) agarose gel. Electrophoresis was performed at 15 V for 5 h and visualized using blue LED transilluminator (Safe Imager 2.0 Blue-Light Transilluminator, Invitrogen).

### 2.8. Immunoblotting Analysis

Whole protein extraction and immunoblotting analysis was performed as previously described [26]. In detail, T98G cell pellets were resuspended in ice-cold RIPA buffer (150 mM NaCl, 50 mM Tris-HCl pH 8.0, 1 mM Triton X100, all from Sigma-Aldrich), supplemented with Complete Mini protease inhibitor cocktail (Roche, Basel, Switzerland). Protein samples were quantified by Qubit fluorimeter, using Protein Assay kit (Invitrogen) following the manufacturer’s instructions. Before loading in SDS-PAGE, protein extracts were boiled in Laemmli sample buffer (2% SDS, 6% glycerol, 150 mM B-mercaptoethanol, 0.02% bromophenol blue, and 62.5 mM Tris-HCl pH 6.8). After electrophoresis, proteins were transferred onto a nitro-cellulose membrane Hybond-C Extra (GE Healthcare, Milan, Italy). Membranes were blocked with 5% nonfat milk in PBS containing 0.1% Tween 20 (*v*/*v*) and incubated over-night at 4 °C with primary antibodies. The primary employed antibodies were: caspase-3 (#D3R6Y), caspase-7 (#D2Q3L), caspase-9 (#C9), Bcl-2 (#2876), Bcl-xL (#54H6), PARP-1 (#46D11), and LC3B (#2775) (Cell Signaling; diluted 1:2000); BACT and α-tubulin (Cell Signaling, #4967 and #2144, diluted 1:6000) were used as internal loading controls. Species-specific peroxidase-labelled ECL secondary antibodies (Cell Signaling, diluted 1:4000) were employed. Protein signals were revealed by the Weststar Supernova Kit (Cyanagen, Bologna, Italy) and visualized using Chemidoc MP system (Biorad, Hercules, CA, USA).

### 2.9. Clonogenic Assay

To evaluate plating efficiency, a clonogenic survival assay was performed as described [27]. Briefly, 24 h after BBR and/or PDT stimulation, cells were trypsinized and seeded into 6-well plates (10^4^ cells per well), incubated for two weeks at 37 °C and then fixed with ethanol. Cells were stained with 0.5% (*v/v*) Crystal Violet (Sigma-Aldrich), and colonies that contained more than 50 cells were automatically counted using ImageJ colony-counter (https://imagej.nih.gov/ij/plugins/colony-counter.html, accessed on 19 April 2020). The number of clones/well was calculated and normalized to the corresponding control samples. Each experiment was performed in triplicate.

### 2.10. Statistical Analysis

The data were analyzed using the statistics functions of the MedCalc statistical software version 18.11.6. (http://www.medcalc.org, accessed on 15 February 2021). The ANOVA test differences were considered statistically significant when *p* < 0.05.

## 3. Results

### 3.1. BBR Effect and Cellular Internalization in Cancer and Normal Astrocyte Cells

According to published results, reduction in astrocytoma cell line viability was observed after 24 h administration of BBR in a range of 20–200 μg/mL [28,29,30]. Moreover, the compound was reported to be efficiently internalized into the cells after 3 h incubation [19]. Here, different BBR concentrations were administered for 4 h and subsequently cells were analyzed at 24 h post treatment (p.t.).

Specifically, by means of MTS assays aimed at assessing cell viability, the cytotoxicity of BBR in different low- (i.e., Res186) and high-grade (T98G, U87-MG, U373-MG, and U138-MG) astrocytoma cells lines was investigated and compared to the effects of this compound in rat primary astrocytes (Figure 1).

As reported, BBR affected astrocytoma cells viability in a dose-dependent manner, while normal astrocyte cells showed only a slight reduction of viability after the alkaloid administration doses. In particular, when treated with 200 μg/mL BBR for 4 h, T98G cells exhibited the lowest average viability (i.e., 19.50%) compared to Res186 (39.44%), U87-MG (44.37%), U373-MG (72.36%), and U138-MG (72.37%); on the contrary, normal rat astrocytes showed 85.13% viability under these conditions.

To evaluate if the cytotoxic effect of BBR in cancer cells was dependent on uptake efficiency, BBR internalization was analyzed by flow cytometry (Amnis ImageStream MkII, Luminex). For this purpose, T98G cells and normal rat astrocytes were treated with 200 μg/mL BBR for 4 h and visualized after 24 h p.t. As reported in Figure 2, normal rat astrocytes displayed a lower percentage of BBR-positive fluorescent cells compared to T98G cells (i.e., 58.32 and 97.03%, respectively). Moreover, the intensity of fluorescence was higher in the astrocytoma cell line.

Flow cytometry analysis (Amnis ImageStream MkII) was then employed to evaluate differences of BBR accumulation in nuclei of T98G cells (Appendix A). Following BBR administration (200 μg/mL for 4 h) and DRAQ5 nuclear staining, the percentage of cells displaying nuclear co-localization signals was 14.78%. Furthermore, in agreement with that previously described by Agnarelli et al. [28], a higher BBR concentration (i.e., 500 μg/mL) significantly increased BBR nuclear internalization, resulting in 29.04% of BBR-positive cells. To shed light on the cytoplasmic localization of BBR in T98G cells, fluorescent and confocal microscopy were used to assess BBR and mitochondria co-localization signals following administration of 200 μg/mL BBR for 4 h. At 24 h p.t, cells showed extensive co-localized signals (Figure 3).

### 3.2. BBR Associated with PDT Induces Apoptosis

BBR cytotoxicity and the combined effect of PDT stimulation was investigated in normal rat astrocytes, low- (Res186) and high-grade astrocytoma cells (i.e., U87-MG, U138-MG, T98G, and U373-MG). Among the investigated cells, the T98G cells displayed at 200 μg/mL BBR the highest cytotoxic PDT induced effect (Figure 4). Based on these results, deeper cellular and molecular investigations were therefore focused on T98G cells.

Since BBR was previously reported to induce apoptotic and autophagy programmed cell death pathways in different cancer cells [28,29,30], a preliminary evaluation of Annexin V and LC3 expression in T98G cells was performed. Apoptosis induction was primarily assayed through cytofluorimetric Annexin V experiments in T98G cells. Cells treated with increasing BBR concentrations (i.e., 20, 50, 100, 200 μg/mL for 4 h) or exposed to a combined BBR and led stimulation (4 min) treatment (BBR + LED) were assayed at 24 h p.t. Untreated (NT) and LED-only treated cells were used as controls (Appendix A). Remarkably, the combined scheme (BBR + LED) resulted in a dose-dependent reduction in the fraction of live cells associated with a significant increase in early and late apoptotic events (in particular, using 200 μg/mL BBR: BBR coupled with LED stimulation = 46.68% total apoptosis; BBR = 29.82%; LED = 9.18%; untreated cells = 7.52%).

The above mentioned BBR and PDT treatment resulted also in affecting T98G long-term viability: at two weeks after a 4-h exposure to 100 or 200 μg/mL BBR following PDT scheme, clonogenic assays showed a dose-dependent decrease in the number of T98G colonies (Appendix A).

In relation to the autophagic pathway, no significant variations in LC3B expression by Muse cytofluorimetric and immunoblotting analysis in the presence of bafilomycin A1 (10 nM) were scored in BBR-led stimulated T98G cells compared to untreated ones (Appendix A).

Microscopy examinations of T98G cells, highlighted marked cytotoxic effect of BBR (200 μg/mL for 4 h) in conjunction with PDT stimulation, characterized by changes in morphology and pronounced cell detachment; furthermore, TEM analysis revealed nuclear damage, cell shrinkage, and blebbing with release of apoptotic bodies (Figure 5).

According to the indication of an apoptotic involvement, flow cytometric analysis of BBR and BBR-PDT treated cells evidenced an increase in nuclear fragmentation processes compared to untreated and LED-only treated controls (Figure 6).

To further confirm the nuclear fragmentation patterns induced by BBR and PDT stimulation, genomic DNA was extracted from untreated, LED-only, BBR-only and BBR coupled with PDT stimulated T98G cells. Gel electrophoresis analysis showed marked DNA fragmentation patterns in BBR and BBR-LED treated cells, compared to untreated and LED stimulated ones (Figure 7).

To assess the activation of apoptosis following BBR administration and PDT stimulation, T98G cells were examined for the expression of the main apoptotic markers. As reported in Figure 8, immunoblotting analysis highlighted a clear decrease in the amount of the examined pro-caspases (i.e., 3, 7, and 9) accompanied by the appearance of the respective cleaved forms, mainly evident in the combined BBR-LED treatment along with the cleavage of PARP-1 full length protein. In addition, a marked reduction in the expression of the anti-apoptotic proteins Bcl-2 and Bcl-xL was reported following BBR administration and with the combined BBR-LED treatment.

Furthermore, two caspase-specific cytofluorimetric assays (i.e., Muse Caspase 3/7 and Muse MultiCaspase, Luminex), reported respectively in Appendix A, clearly documented the marked and significant increase in caspase activation following the blue-LED photo-activation of BBR in T98G cells, accompanied by a drastic reduction in viable cells. Of note, according to the reported scores, LED stimulation alone did not affect cell viability.

To further verify the involvement of the apoptotic process following PDT stimulation, the experimental scheme of BBR administration in presence/absence of photo-activation was combined with the provision of the specific synthetic peptide Z-VAD(OMe)-FMK, a cell permeable, irreversible pan-caspase inhibitor that displays non-cytotoxic effects [31]. Specifically, the peptide was administered at 100 μM, 4 h after BBR administration and/or PDT stimulation. T98G cells were then evaluated after 24 h p.t., in viability and in caspase expression using the cytofluorimetric Muse MultiCaspase assay (Luminex). As illustrated in Figure 9, the pan-caspase inhibitor significantly rescued cell viability to 63.20% compared the corresponding treatment (BBR and PDT, without Z-VAD(OMe)-FMK) whose overall viability declined to 5.15%. An opposite and significant trend was scored considering caspase+/dead cell events (58.40 *vs* 18.75%, respectively).

To better discern the effect of the photo-activation scheme within T98G cells, the cytometric Muse Oxidative Stress assay (Luminex) was employed to simultaneously determine the count and percentage of cells undergoing oxidative stress based on the intracellular detection of superoxide radicals (ROS) using dihydroethidium, a well-characterized reagent extensively used to detect ROS [31]. As schematized in Figure 10, ROS positive cells in the combined scheme evaluated at 24 h p.t. reached 85.13%, compared to 37.90% for BBR administration (200 μg/mL for 4 h) and to 18.53% in untreated and 24.63% in LED-only stimulated cells.

Next, to evaluate differences in the sub-cellular localization of BBR, particularly within mitochondria, following PDT stimulation, fluorescent microscopy and flow cytometry (Amnis ImageStream MkII, Luminex) examinations were used to visualize and quantify BBR and mitochondria co-localization signals following BBR administration (200 μg/mL for 4 h) with/without LED stimulation and finally analyzed at 24 h p.t. The results, reported in Appendix A, indicated roughly similar co-localization ratios between the investigated samples (i.e., 2.424 and 2.396, respectively for BBR and BBR and LED combined scheme), suggesting that rather than differences induced by LED stimulation in the subcellular localization of BBR, photochemical processes might be responsible for the more pronounced cytotoxic effect produced by the combination of BBR and light stimulation.

Then, since ROS over-production leads to mitochondria dysfunction as widely documented [32,33], the functionality of mitochondria following BBR and PDT stimulation was evaluated in T98G cells, using the specific Muse MitoPotential cytofluorimetric assay (Luminex). As a result, a prominent mitochondrial depolarization was scored in the combined treatment scheme, with in particular a highly significant increase in both mean death/depolarized and in total depolarized cells (52.80 and 70.32%, respectively), compared to BBR (10.98 and 25.62%), LED treated (18.00 and 29.34%) and untreated cells (16.61 and 22.24%) (Figure 11). Again, LED-only stimulation exhibited similar mitochondria depolarization percentages of untreated cells. Furthermore, BBR and the combined treatment with PDT caused significant mitochondrial damage compared to untreated and LED-only treated cells.

To confirm mitochondrial damage and the subsequent activation of caspase pathways, a flow cytometry quantification of the release of Cytochrome c from the mitochondria of apoptotic cells was performed using Amnis ImageStream MkII. As reported in Figure 12, untreated (NT) or LED-only stimulated cells (LED) demonstrated higher levels of Cytochrome c staining while BBR-treated (BBR) and BBR-LED stimulated (BBR + LED) cells which have released their Cytochrome c from the mitochondria to the cytoplasm showed significant reduced staining intensity when probed with an anti-Cytochrome c DyLight 650 conjugated antibody.

Lastly, ultrastructural analysis by TEM, reported in Figure 13, revealed swollen mitochondria with clear damage to their cristae and a reduction in the mitochondrial matrix density in correspondence with BBR and BBR LED-combined treatments.

## 4. Discussion

Among glial-derived cancers, GBM is the most aggressive and common adult primary intracranial neoplasm with extremely poor prognosis. Conventional treatments for GBM, as surgery, radiotherapy, and chemotherapy, produced relatively modest effects towards the patients’ prognosis. Consequently, new therapeutic strategies are nowadays urgently required [4]. In this regard, increasing evidence has demonstrated that PDT, a two-stage treatment that combines light energy with a drug (photosensitizer), is significantly effective against different types of cancers [34], including malignant brain tumors [8,9,35,36]. Importantly, in the search for novel photosensitizing agents, the natural plant-derived alkaloid BBR has demonstrated a broad spectrum of efficacy [37,38,39] as well as in conjunction with PDT anticancer schemes [19].

To test the potentiality of this compound in PDT applications in GBM cells, internalization studies were first performed to confirm that BBR tend to accumulate at higher concentrations in the tumor compared to normal cells, possibly due to the BBR affinity to low density lipoproteins (LDL) that act as carriers of BBR moieties [19]. In tumor cells, indeed, the increase of cholesterol catabolism results in an over-expression of LDL receptors (specifically, B/E receptors). It was previously reported that LRP are abundant in different GBM cell lines [40,41,42]. For example, in U87-MG glioma cells, Andreazza and collaborators [19] estimated the binding of 400 BBR molecules per LDL moiety. It has been documented that photosensitizers, with cationic charges as BBR, can localize in mitochondria [43] due to the influence of the mitochondrial membrane potential as well as the lipid bilayer composition of the membrane [44]. It has also been assayed that carcinoma cells preferentially accumulate and retain certain cationic dyes to a much greater extent than most normal cells [45]. This evidence might therefore explain the significant difference we scored in BBR cellular uptake, comparing normal rat astrocytes with T98G GBM cells. Furthermore, in agreement with Lin et al. [46], a nearly 100% uptake was observed when tumor cells were incubated with BBR for a time interval between 2 and 4 h.

Different studies reported that BBR mostly affected the mitochondrial function of cancer cells, due to its selective accumulation in these organelles, through the interaction with the adenine nucleotide translocator [47]. On the other hand, we confirmed in T98G cells that a relatively high BBR concentration (i.e., 500 μg/mL) induced a significant accumulation of the alkaloid even in the nuclei compartments, as also documented in the U343 GBM cell line [29]. Importantly, the intracellular distribution of PS, including therefore BBR, is one of the key determinants of PDT efficacy, and its final subcellular localization is well correlated with the specific induced mechanism of cell death [48,49]. We therefore employed a specific BBR concentration evaluating through cytofluorimetric analysis its prevalent accumulation within the cytoplasm of T98G cells. Next, considering that a light device with a blue LED source produced an excited state of BBR [50], we employed a similar energy stimulation to evaluate the PDT effect in T98G cells, one of the most employed established GBM cell lines. Furthermore, as documented by Caverzan and collaborators [51] among several tested GBM cell lines, T98G displayed the higher gene-transcription levels of antioxidant enzymes accompanied by the smallest endogenous oxidative-stress levels; these cells can be therefore reasonably considered as paradigmatic substrates characterized by a relatively high level of intrinsic resistance to the photo-activation process.

It was previously reported that BBR treatment activated apoptosis enhancing procaspase-3 cleavage and therefore caspase-3 activity in several tumor cells [28,52,53,54], including U251 and U87-MG GBM cell lines [28,30]. Conversely, Floriano and collaborators [55] did not report significant differences in the activation of the key executioner of apoptosis caspase 3 in cervical carcinoma cells, leading to the hypothesis that the effect of BBR on key caspase activity may be cell specific. Regarding these considerations, we primarily confirmed in vitro evidence reporting that BBR induced apoptosis of T98G cells by means of the activation of mitochondrial/caspase-dependent pathways in agreement with previous results [28,56]. As initial hallmarks of the apoptosis involvement, we highlighted genomic DNA and nuclei fragmentation patterns, subsequently accompanied by caspase activation through immunoblotting and cytofluorimetric independent assays. Importantly, we also reported a significant reduction in the expression of two main anti-apoptotic proteins, i.e., Bcl-2 and Bcl-xL. In this regard, it is important to underline that the reduction of Bcl-2 through the use of specific antagonists, represents a potential strategy for the treatment of tumors [57]. In addition, we reported a massive increase in intracellular production of ROS and of mitochondrial membrane depolarization events accompanied by a significant release of mitochondrial Cytochrome c following BBR and light stimulation. Electron microscopy analyses documented that BBR and the combined treatment with photo-stimulation produced a marked damaged of mitochondria, with disrupted and discontinuous outer membranes and cristae. Again, the identification of an evident mitochondria vulnerability might be a promising direction for the generation of antitumor drug development [58].

Of importance, according to our documented results, these apoptotic features were further amplified in T98G cells following the photosensitizing scheme, while the control treatment (i.e., with LED-only stimulation) was not effective and nearly identical to the untreated samples. Particularly, through the use of a specific pan-caspase inhibitor, i.e., Z-VAD(OMe)-FMK already assayed in T98G cells [59], we further confirmed the involvement of the apoptotic process within the induced cytotoxicity in T98G cells. This overall evidence strongly supported the concept that BBR can constitute a valid substrate in PDT stimulation schemes. Additional results highlighted similar cytotoxic effect of the developed BBR-PDT protocol in low- (Res186 and Res 259) and high-grade (U87-MG, U373-MG, and U138-MG) glioma cells.

On the other hand, our investigation did not highlight a marked involvement of the autophagy process following BBR and irradiation. First, direct optical microscope examinations did not reveal the presence of cytoplasmic vacuoles resembling autophagosomes and specific cytofluorimetric and immunoblotting analysis of LC3 expression, one of the most validated autophagy markers according to current guidelines [60], indicated slight but not significant variations following the combined BBR-LED treatment. Differently, other contributions reported autophagy activation in response to photodynamic stimulation at 24 h p.t., albeit in cervical carcinoma cancer cells [56]. On the other hand, the autophagy activation was highlighted in U343 GBM cells but in the absence of caspase-3-dependent apoptosis [16]. However, we cannot exclude that the activation of the autophagy process might be strictly related to the BBR concentration adopted and to the physical parameters of the irradiation scheme. It is also important to consider that the crosstalk between autophagic and apoptotic signaling pathways is complex and not always discernible in their specific contribution. A detailed study reported that an autophagic survival response was triggered in the presence of a relatively low level of photo-damage and that the programmed cell death fate was rescued when apoptosis was blocked [61]. This experimental scenario might be similarly consistent to that revealed in our study. On another level of complexity, some chemotherapeutics but not photosensitizers per se, induced both pathways, namely temozolomide in established glioma cells [62] and givinostat in glioma stem cells [63]. Consequently, in our specific experimental scheme, based on the synergic effect of BBR and PDT, additional studies might be necessary to better decipher at the molecular levels the players involved in the apoptotic/autophagy interplay.

In relation to GBM, it was recently reported that *IDH1* mutation and the related production of 2-HG can induce oxidative stress, autophagy, and apoptosis [64]. Based on these premises, it might be interesting to evaluate if *IDH1* mutated cells, for instance R132H polymorphism, induces additional and cumulative PDT effects (i.e., production of ROS and programmed cell death induction pathways) compared to cells exhibiting the *wild-type* allele. As an additional perspective, in vivo studies are scheduled to determine the possibility that BBR combined with PDT and combined with optical nanofiber devices could be an effective therapeutic agent for the management of malignant gliomas.

## Figures and Tables

**Figure 1 jpm-11-00942-f001:**
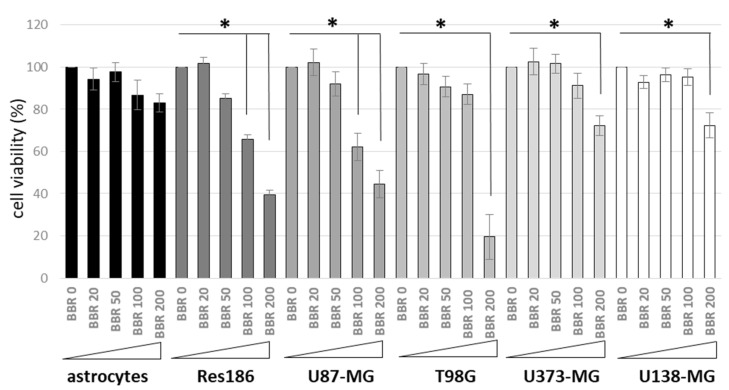
Cell Viability Assays after BBR administration. Low- (i.e., Res186) and high-grade (U87-MG, T98G, U373-MG, and U138-MG) astrocytoma cells lines and normal rat primary astrocytes were treated with different concentrations of BBR (0, 20, 50, 100, 200 μg/mL for 4 h). Colorimetric MTS analysis was carried out after 24 h p.t. All experiments were performed in triplicates with independent assays. Asterisks indicated statistical significance compared to untreated cells (*p* < 0.05, Anova One-way).

**Figure 2 jpm-11-00942-f002:**
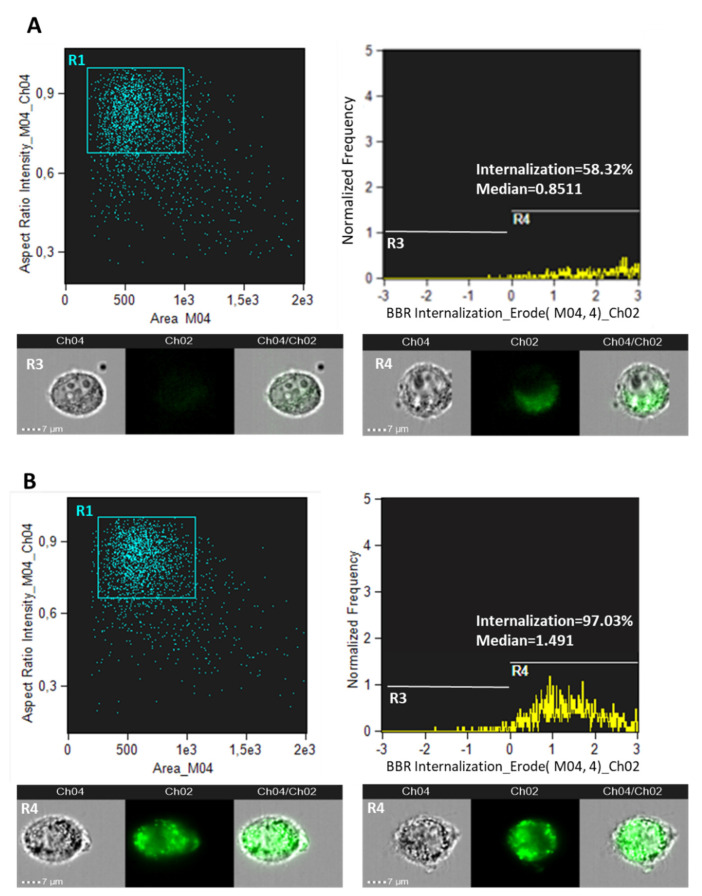
BBR cellular internalization. Amnis ImageStream MkII flow cytometry analysis of BBR (200 μg/mL for 4 h) internalization in normal rat astrocytes (**A**) and in T98G cells (**B**) analyzed after 24 h p.t. Trypsinized cells were analyzed for bright field (Ch04) and for BBR fluorescence (Ch02) at 60× magnification. Percentages of focused cells with BBR internalization spots and Internalization Erode median parameters are reported. BBR internalization was analyzed by “Internalization” wizard with the following gating strategy: single cells were gated (using Area/Aspect Ratio Intensity, R1); R1 was then gated for focused cells using the “Gradient-RMS” feature and finally Ch2 intensity was measured with “Internalization” feature and an “Erode” mask applied. R2 gate (not shown) identified focused cells. Examples of cell images of R3 (negative control, i.e., cells without BBR) and R4 gates are shown.

**Figure 3 jpm-11-00942-f003:**
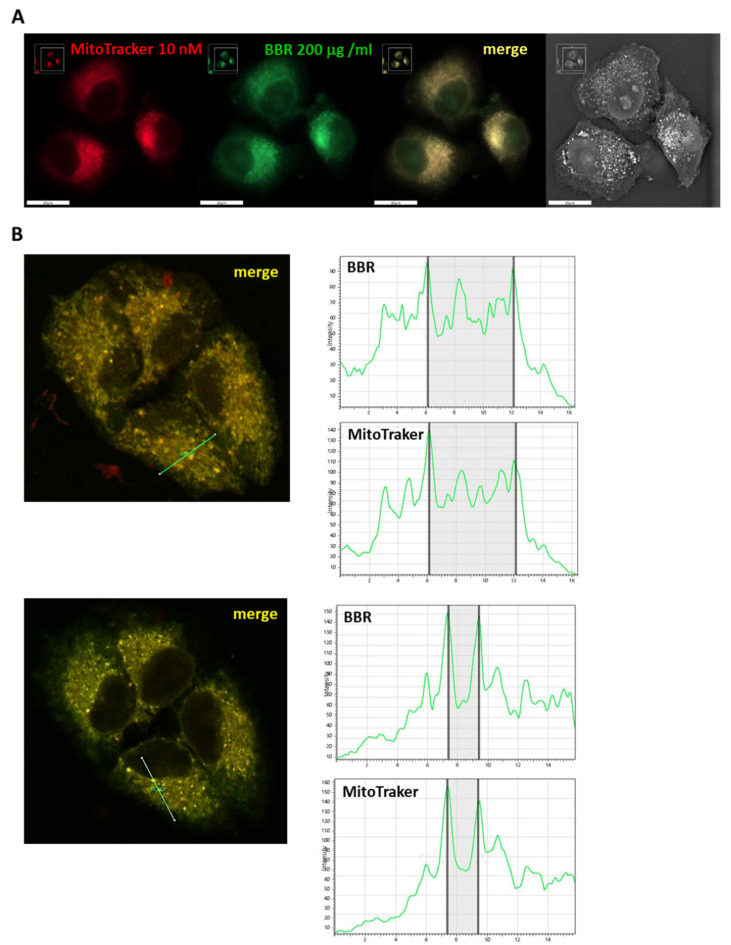
BBR and mitochondria co-localization. T98G cells were incubated for 4 h with BBR (200 μg/mL) and analyzed after 24 h p.t. Before microscopy examination, cells were treated with 10 nM MitoTracker deep Red (Thermofisher) for 45 min and then visualized using a Nanolive microscope, scale bars = 20 μm (**A**). Co-localization analysis was performed using confocal microscopy, reporting histograms of BBR and MitoTracker profiles (**B**).

**Figure 4 jpm-11-00942-f004:**
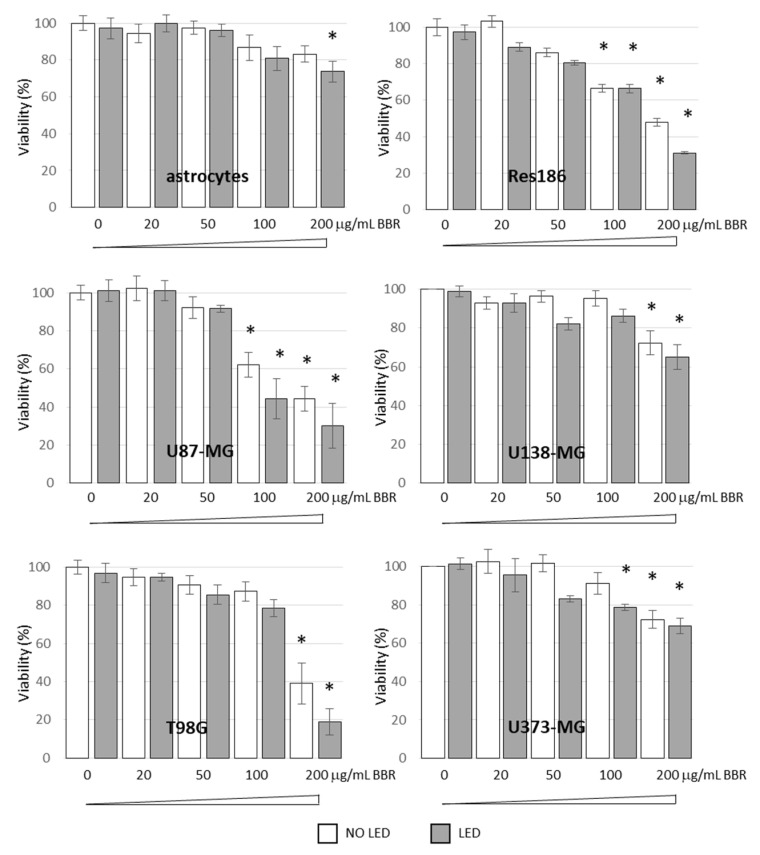
Cell Viability Assays after BBR and/or PDT (LED) administration. Normal rat astrocytes, low- (i.e., Res186) and high-grade (U87-MG, T98G, U373-MG. and U138-MG) astrocytoma cell lines were treated with different concentrations of BBR (0, 20, 50, 100, 200 μg/mL for 4 h, white bars) and exposed to LED stimulation for 4 min (grey bars). Colorimetric MTS analysis was carried out after 24 h p.t. All experiments were performed in triplicates with independent assays. Asterisks indicated statistical significance compared to untreated cells (*p* < 0.05, Anova One-way).

**Figure 5 jpm-11-00942-f005:**
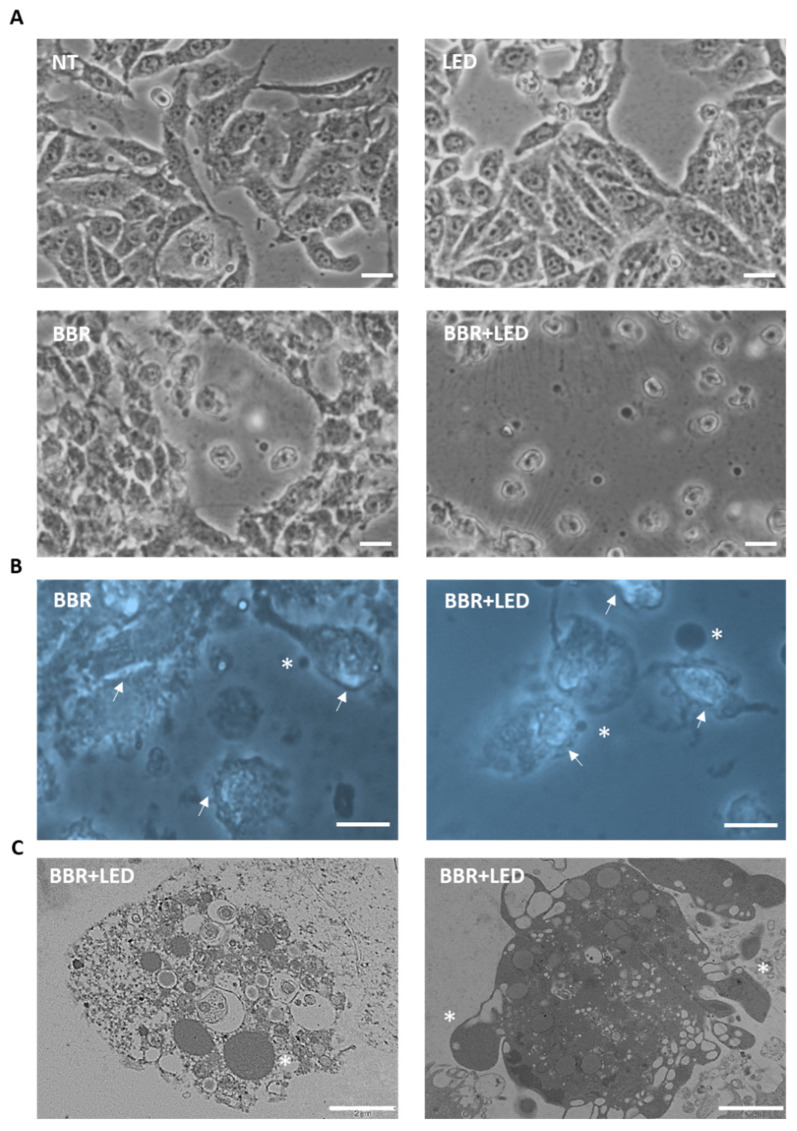
Cytotoxic effect of BBR and PDT. Phase contrast (**A**), fluorescence (**B**) and transmission electron microscopy (**C**) analysis of untreated T98G cells (NT), with only LED stimulation, after BBR administration (200 μg/mL for 4 h) and with the combined BBR + LED treatment. Cells were analyzed after 24 h p.t. BBR intracellular accumulation (ext. 355 nm) is indicated by arrows (**B**); asterisks indicated apoptotic bodies (**B**,**C**). Scale bars = 10 μm (**A**,**B**) and 2 μm (**C**).

**Figure 6 jpm-11-00942-f006:**
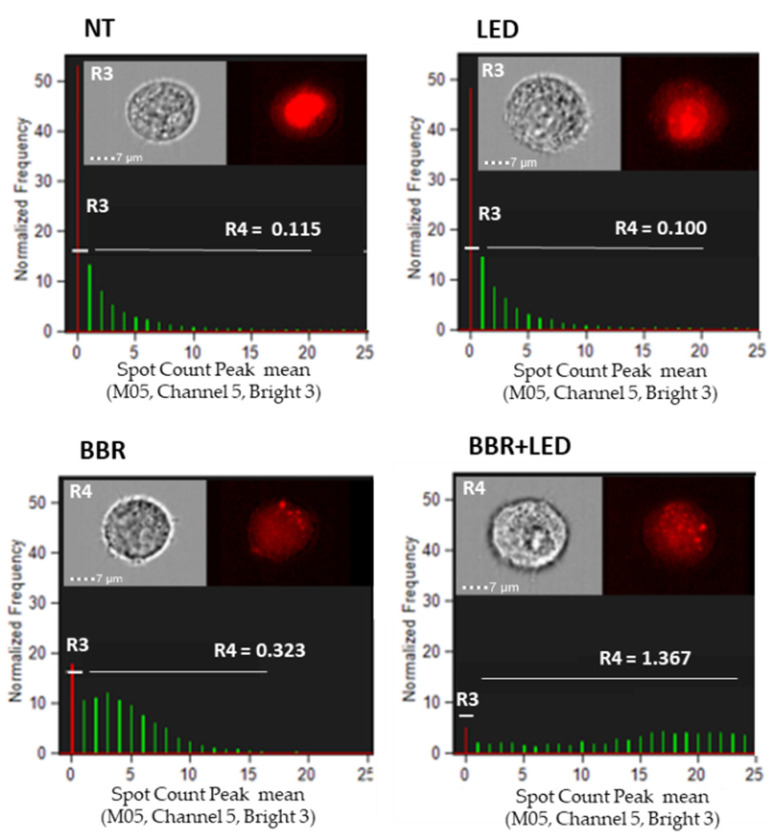
Nuclear fragmentation analysis. Amnis ImageStream MkII flow cytometry analysis of nuclear fragmentation in T98G cells as untreated (NT), exposed to LED stimulation (LED), BBR (200 μg/mL for 4 h) administration and with the combined scheme (BBR + LED), evaluated after 24 h p.t. Trypsinized cells were analyzed for bright field (Ch04) and for nuclear fluorescence (Ch05) using DRAQ5 dye (Invitrogen) at 60× magnification. R3 and R4 gates indicated integrity and nuclear fragmentation signals, respectively. Mean Spot Count Peak values (i.e., 0.323 and 1.367) and cell images representative of R3 and R4 gates are reported. R1 and R2 gates (not shown) identified focused cells.

**Figure 7 jpm-11-00942-f007:**
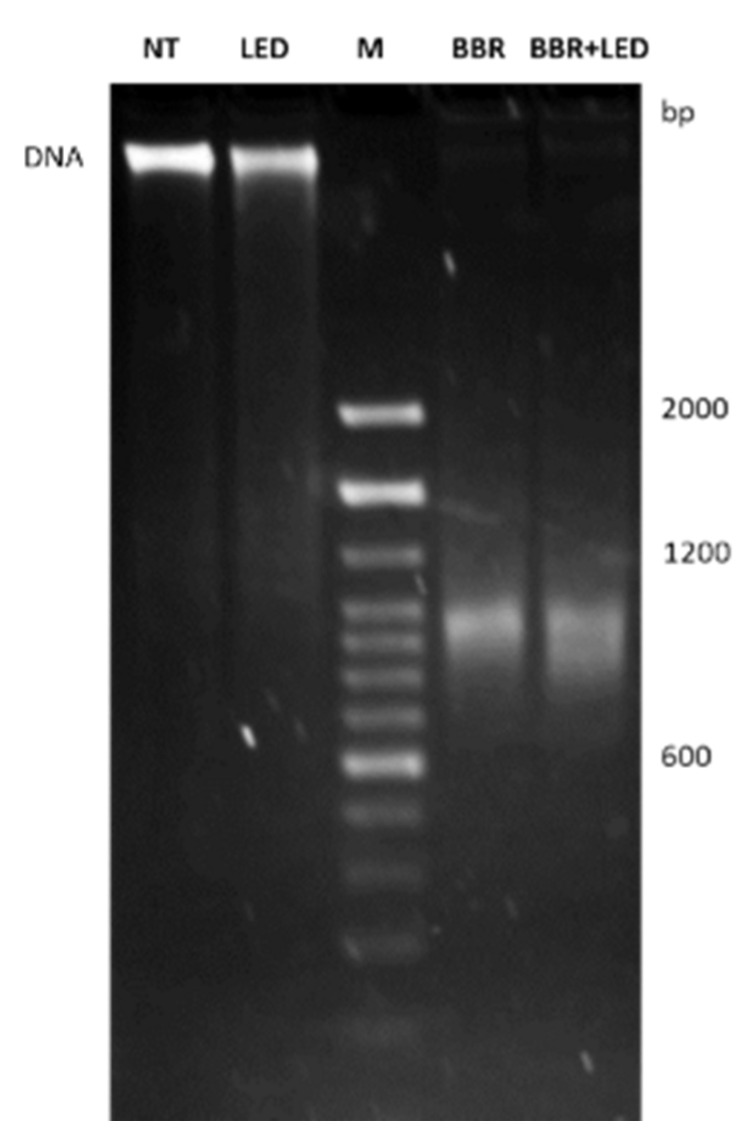
Genomic DNA analysis following BBR and LED stimulation. Agarose gel electrophoresis analysis in T98G cells as untreated (NT), exposed to LED stimulation (LED), BBR (200 μg/mL for 4 h) administration and with the combined LED+BBR treatment evaluated at 24 h p.t. After cells trypsinization, genomic DNA was extracted and quantified using Qubit DNA HS kit (Invitrogen): 180 ng of each DNA was loaded onto a 2% agarose gel and visualized after 5 h running at 25 V. M is the 100 bp DNA ladder (Invitrogen).

**Figure 8 jpm-11-00942-f008:**
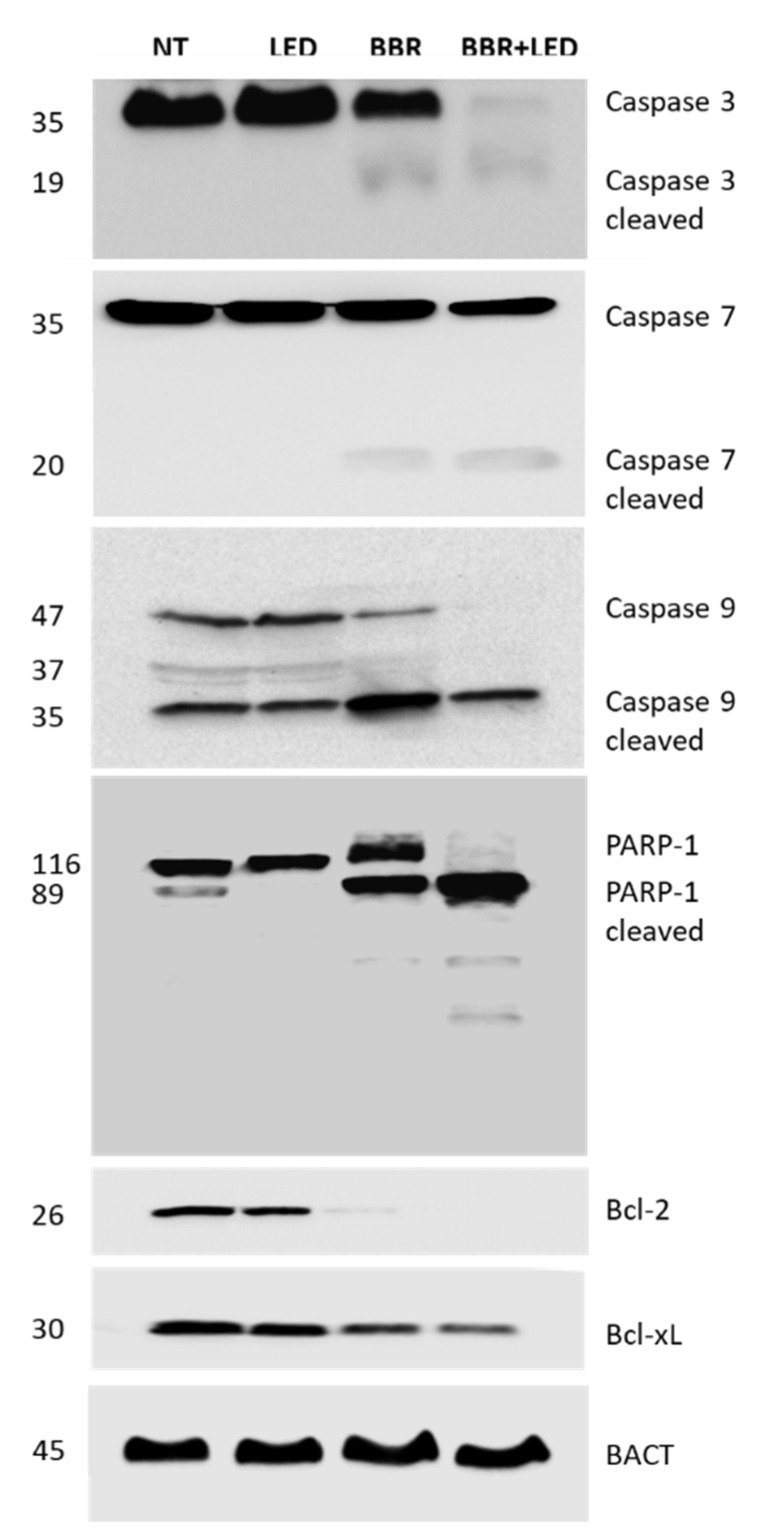
Expression of apoptotic markers. Protein expression in untreated (NT), LED stimulated (LED), BBR (200 μg/mL for 4 h) treated, and BBR + LED exposed T98G cells were evaluated after 24 h p.t. by immunoblotting. For each sample, 40 μg of total protein extracts was loaded into 10% (for PARP-1) and 12% polyacrylamide gels. All the experiments were performed in triplicate with independent assays. A representative experiment is shown. Molecular weights in KDa are reported.

**Figure 9 jpm-11-00942-f009:**
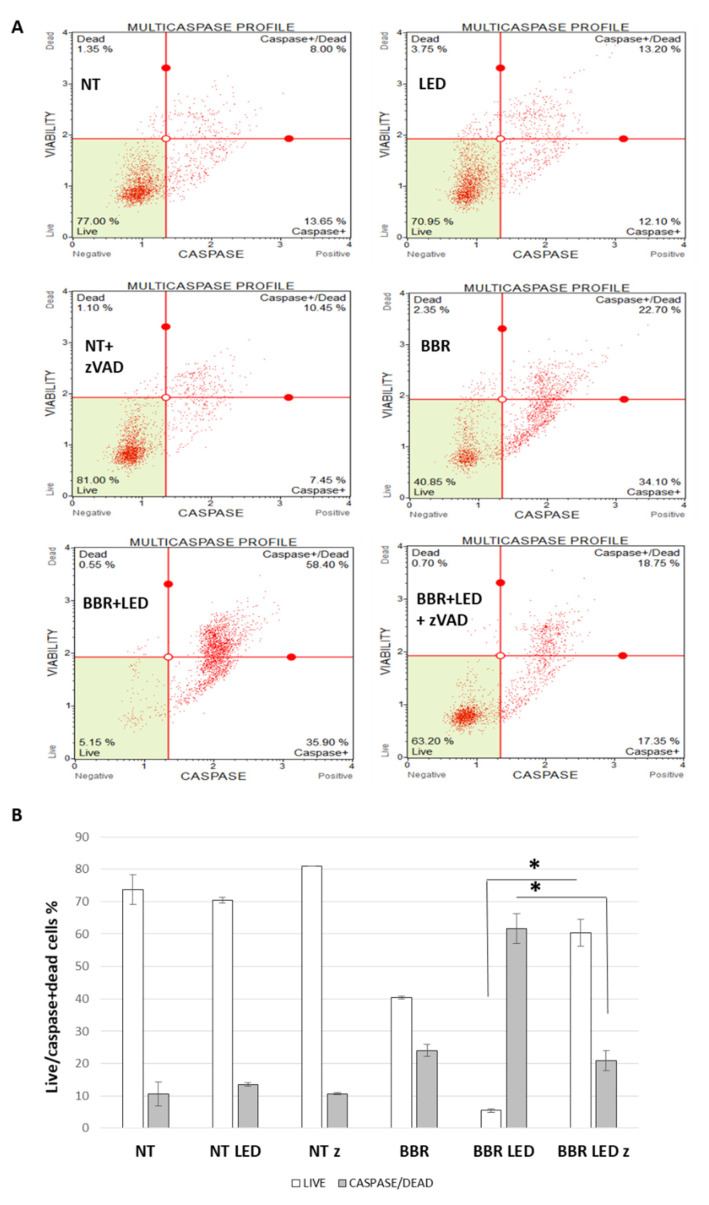
Caspase cytometric analysis of BBR and PDT scheme following a pan-caspase interference. Cytofluorimetric analysis of caspase activation in T98G cells as untreated (NT), exposed to LED stimulation (LED), BBR (200 μg/mL for 4 h) administration, with the combined LED+BBR after and in presence of 100 μM z-VAD(OMe)–FMK inhibitor of pan-caspases, evaluated at 24 h p.t. (**A**). Trypsinized cells were analyzed for apoptosis induction using the Muse MultiCaspase kit (Luminex). The histogram of the live/dead cells percentages of two independent experiments is reported (**B**). Asterisks indicated statistical significance of live and caspase/dead BBR + LED with inhibitor (z), compared to BBR + LED treated cells (*p* < 0.05, Anova One-way).

**Figure 10 jpm-11-00942-f010:**
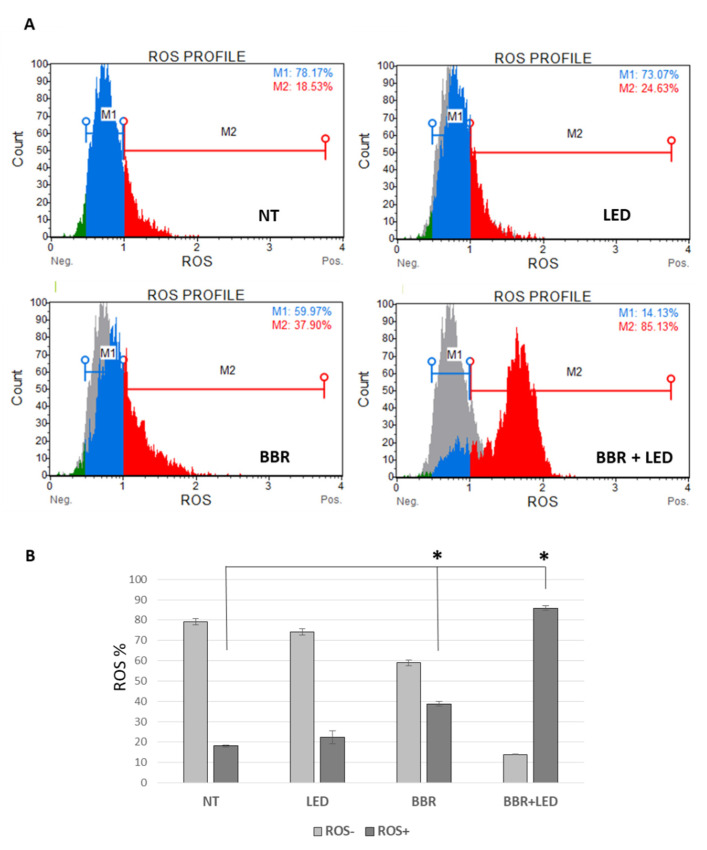
Oxidative stress analysis. Cytofluorimetric analysis of Reactive Oxygen Species (ROS) in T98G cells as untreated (NT), exposed to LED stimulation (LED), BBR (200 μg/mL for 4 h) administration and with the combined LED+BBR, evaluated after 24 h p.t. (**A**). Trypsinized cells were analyzed for count and percentage of cells undergoing oxidative stress based on the intracellular detection of ROS using the Muse Oxidative Stress kit (Luminex). The histogram of the averages of three replicas of ROS- (M1 gate) and ROS+ (M2) cells percentage is reported (**B**). Asterisks indicated statistical significance compared to untreated cells (*p* < 0.05, Anova One-way).

**Figure 11 jpm-11-00942-f011:**
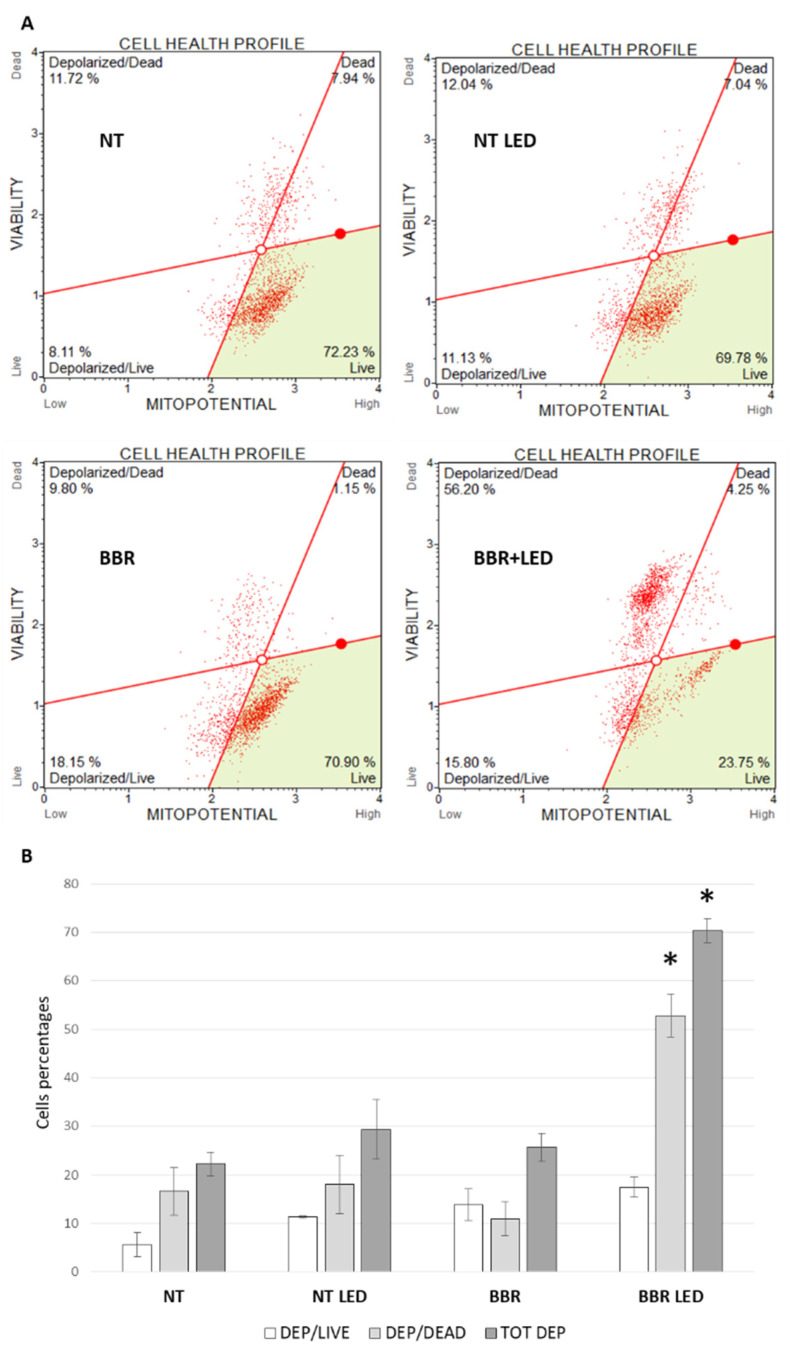
Mitochondria depolarization analysis. Cytofluorimetric analysis of mitochondria depolarization in T98G cells as untreated (NT), exposed to LED stimulation (LED), BBR (200 μg/mL for 4 h) administration and with the combined LED+BBR scheme, evaluated after 24 h p.t. (**A**). Trypsinized cells were analyzed for count of live and dead depolarized percentage of cells using the Muse MitoPotential kit (Luminex). The histogram of the averages of two replicas of cells percentage is reported (**B**). Asterisks indicated statistical significance compared to untreated cells (*p* < 0.05, Anova One-way).

**Figure 12 jpm-11-00942-f012:**
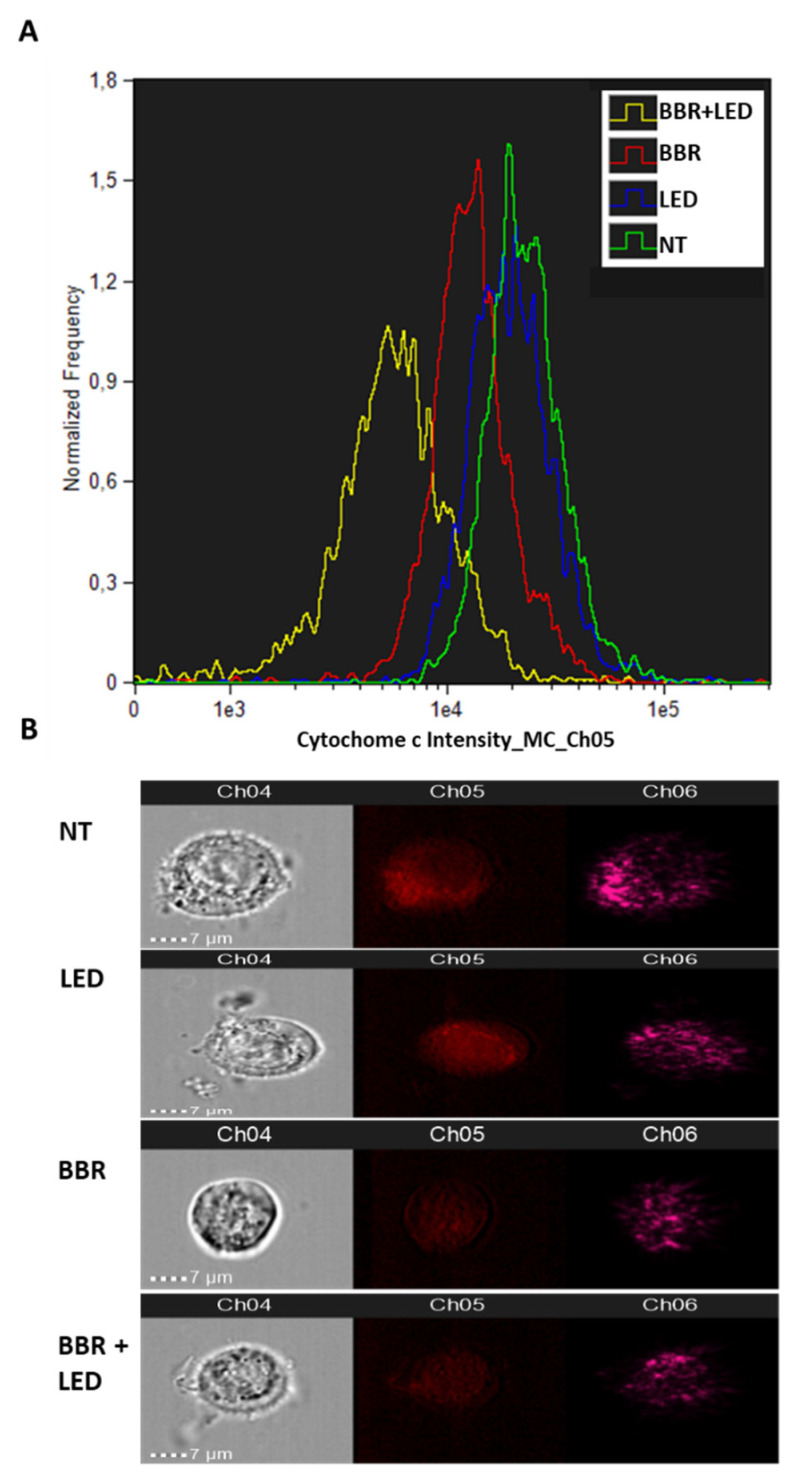
Cytochrome c analysis. Amnis ImageStream MkII flow cytometry analysis of Cytochrome c expression in T98G cells as untreated (NT), exposed to LED stimulation (LED), BBR (200 μg/mL for 4 h) administration and with the combined BBR + LED (**A**), analyzed after 24 h p.t. Trypsinized cells were analyzed for bright field (Ch04) and for Cytochrome c fluorescence (Ch05) at 60× magnification. Intensity plot of Ch05 is reported (A) with the following average values: NT = 23,185; LED = 21,808; BBR = 13,042; BBR + LED = 5394. Representative cell images are reported (**B**). Ch06, scatterplot.

**Figure 13 jpm-11-00942-f013:**
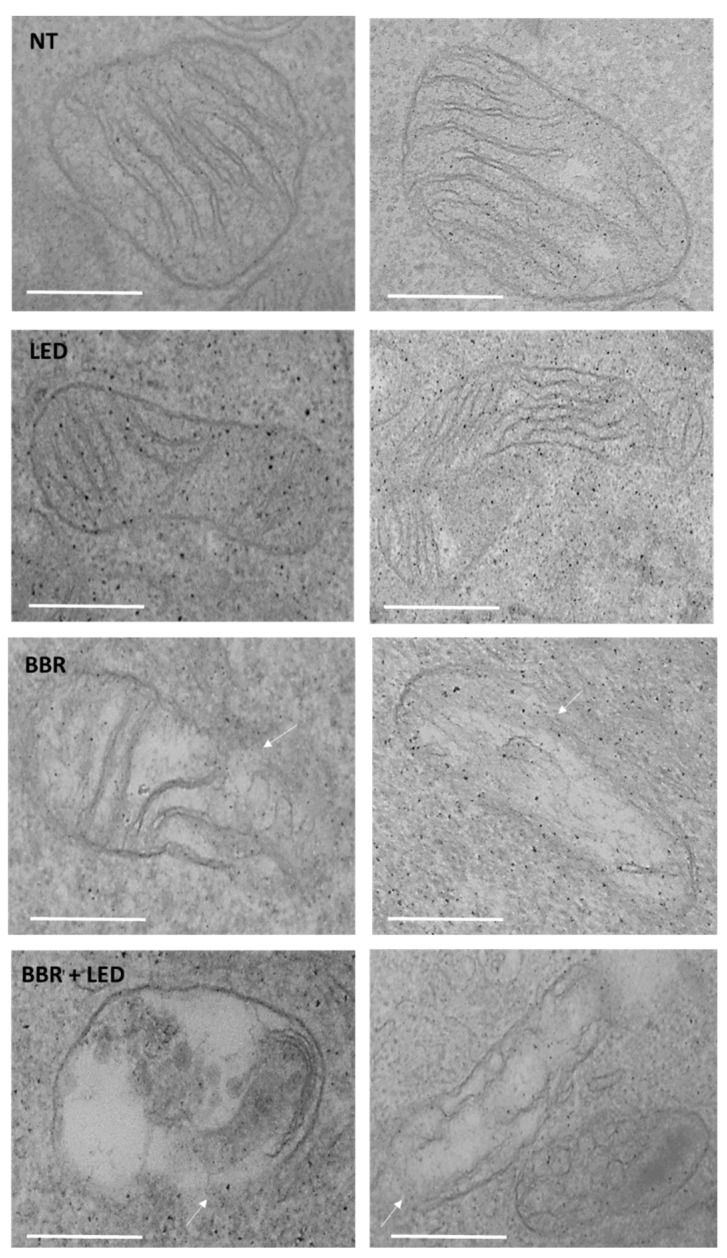
Ultrastructural analysis of mitochondria. T98G cells as untreated (NT), exposed to LED stimulation (LED), BBR (200 μg/mL for 4 h) administration, and with the combined BBR + LED scheme, were trypsinized and analyzed by TEM after 24 h p.t. Arrows indicated mitochondrial membranes damages. Scale bars (0.5 μm) are reported.

## Data Availability

Not applicable.

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
