# Peer review of "Berberine Photo-Activation Potentiates Cytotoxicity in Human Astrocytoma Cells through Apoptosis Induction"

_jpm, 2021, doi:10.3390/jpm11100942_

Round 1

Reviewer 1 Report

Dear Authors:

The manuscript is better. I have no other comments.

Author Response

Thank you

Reviewer 2 Report

The manuscript form Francesca Carriero and colleagues entitled „Berberine Photo-activation Potentiates Cytotoxicity in Human Astrocytoma Cells Through Apoptosis Induction“ deals with an interesting treatment approach for glioblastoma (i.e. combined use of berberine and blue light transilluminator).

In it´s present form the manuscript is not suitable for publication. The main concern are some technical problems and the redundance of data recapitulating the same observed effects, which makes it difficult for the reader. It is recommended to streamline the manuscript and to shift some results/data to supplementary material.

Major critics

Introduction

The authors state that berberine shows a reduced toxicity towards normal cells. The accompanying citation #14 is not correct since no data concerning reduced cytotxicity against normal cells are reported. The same accounts for citations 15 and 16….

It is recommended to explain in more detail why berberine is prefrentially internalized by cancer (i.e. glioblastoma) cells. In their discussion section the authors mentioned that LDL bind 400 berberine molecules. In addition, there are reports that glioblastoma cells abundantly express LDL receptors.

Material and Methods/Result section

When reading the manuscript it appears that the authors did not include mock controls. In this regard, the authors should provide data, which include samples in which the BBR- and caspase-inhibitor-treated cells (reagents were solved in DMSO) are compared to controls, which were incubated with the same concentrations of DMSO.

Concerning figures where cytofluorimetric analysis are depicted some compensation problems are obvious (especially in Fig. 10, 11 and 12). Please, revise.

Fig. 12: Shows very defensive gating on M2. There is always some ROS in normal cells. The M2 gate/marker should be shifted to the left, and then it became clear that BBR+LED treatment is very effective.

Discussion

Please, give a comment on the potential route of application for BBR when treating GBM.  

Author Response

All responses are in the PDF file. Please check. Thank You

This manuscript is a resubmission of an earlier submission. The following is a list of the peer review reports and author responses from that submission.

Round 1

Reviewer 1 Report

In the manuscript titled "Berberine Photo-activation Potentiates Cytotoxicity in Human Astrocytoma Cells Through Apoptosis Induction" prepared by Carriero et al., the author investigated photodynamic therapy (PDT) to malignant glioma cell line. They monitored apoptosis pathway, ROS production, and mitochondria changes in glioma cell lines with PDT treatment. I have several concerns as follow:

  1. One of the major concerns is the possible application of PDT in real-world patient care. The present study uses in vitro models and applied blue light stimulation as a model of PDT. However, glioma is an intracranial tumor, which located in the skull with a highly complicated anatomical structure. How could PDT be used in patient care? Is there preclinical animal work currently ongoing to test the in vivo use?
  2. I am skeptical about how the BBR signal is recorded in the present study. It seems there is no other fluorophore conjugated to this compound (e.g., FITC or Alexa Fluor 488), whereas its distribution is monitored through fluorescence-based microscopy/flow cytometry. Could the author be more specific on how BBR is visualized?
  3. Glioma is a heterogeneous disease. Several recent studies showed that glioma with IDH mutation exhibits stronger oxidative stress (e.g., PMID: 32825279, PMID: 32291392, and PMID: 24150401). As one of the mechanisms shown in this study is ROS production, the author may want to include a discussion about the possible use of PDT in this subcluster of glioma.
  4. For the Amnis ImageStream flow cytometry analysis, the author could modify the title of the x/y-axis, so that it would be clear about which parameters were used.
  5. In Figure 2, the R1 gating seems very different between experiments. I think this is not appropriate.
  6. In Figure 3, the colocalization of BBR and MitoTracker is not conclusive. Could the author use a confocal microscope to better visualize the organelle?

Reviewer 2 Report

The authors described an interesting topic of photodynemic therapy (PDT) as an potential treatment for malignant gliomas. They reported a complete study in vitro. This study demonstrated that BBR is an efficient photosensitizer agent and its association with PDT may have anti-cancer ability in malignant gliomas.

There are some suggestions:

  1. Please add more citations of this Journal.
  2. English editing/revision is required.